# Lessons from the deployment and management of public handwashing stations in response to the COVID-19 pandemic in Kenya: A cross-sectional, observational study

Josphat Martin Muchangi[1], Dennis Munai[1]*, Rogers Moraro[1], Astrid Hasund Thorseth[2], Viola Tupeyia[1], Judy Muriithi[1], Jennifer Lamb[2], Richard Gichuki[1], Katie Greenland[2], Sheillah Simiyu[3]

1 Population Health and Environment Department, Amref Health Africa, Nairobi, Kenya, 2 Environmental Health Group, Department for Disease Control, London School of Hygiene & Tropical Medicine, London, United Kingdom, 3 Urbanization and Well-being Unit, Africa Population Health and Research Centre, Nairobi, Kenya

* dennis.munai@amref.org

**Editor:** D. Daniel, Gadjah Mada University Faculty of Medicine, Public Health, and Nursing: Universitas Gadjah Mada Fakultas Kedokteran Kesehatan Masyarakat dan Keperawatan, INDONESIA

## Abstract

During the COVID-19 pandemic in Kenya, 5,311 handwashing stations were distributed by the National Business Compact Coalition (NBCC) to help combat the virus. This study evaluated 316 of these stations across five counties, assessing functionality, usability, and accessibility. Quantitative data, including spot checks and surveys, revealed that 83.9% of the evaluated stations were functional, with paid caretakers, which is associated with higher functionality rates. Qualitative insights underscored challenges such as inadequate signage and limited soap and water availability, particularly affecting individuals with disabilities. Despite initial success, only 61% of stations remained functional 6–8 months post-distribution, often due to relocation by caretakers. Future distributions should prioritize long-term support for caretakers to sustain station functionality. This study highlights the importance of ongoing monitoring and support for public handwashing facilities in pandemic response efforts.

## Introduction

On 13th March 2020, Kenya confirmed its first case of COVID-19. The person identifies was a traveler who had arrived from London a week earlier (1). By the 24th of November 2022, Kenya had recorded 341,043 cumulative confirmed COVID-19 cases and 5,684 cumulative deaths [1].

At the onset of the pandemic, a multisector task force, the National Emergency Response Committee (NERC) comprising health, security, education, transport, finance, and trade sectors, was set up to coordinate the COVID-19 response [2]. NERC identified 7500 COVID-19 transmission 'hotspots' in public settings across Kenya [3]. These public setting hotspots were broadly categorized as any place accessible for use by anyone in the community, that is,

**Data Availability Statement:** Please find below the contact information for the designated institutional body responsible for handling data access requests: • Institution Name/Department: Amref Ethics and Scientific Review Committee • Contact Person's Name: Mr. Samuel Muhula • Contact Person's Position: Senior Manager, Learning and Impact • Contact Person's Email: Samuel. muhula@amref.org • Contact Person's Phone Number: +254 721958734 This contact information is provided to facilitate communication and access to the data used in our study. We assure you that the designated contact person is authorized to handle data access requests on behalf of our institution and will respond promptly to inquiries regarding data availability and access. Please feel free to contact us if you require any further information or clarification regarding the provided contact details.

**Funding:** The author(s) received no specific funding for this work.

**Competing interests:** The authors have declared that no competing interests exist.

common areas for social interaction and/or economic exchange. The areas included transport hubs, shopping centers, markets, places of worship, high-foot traffic streets, public buildings and government offices.

To mitigate the spread of COVID-19 within the country, NERC implemented several non-pharmacological interventions namely: school closures, isolation for suspected cases, contact tracing, a countrywide dusk-to-dawn curfew, closure of non-essential businesses, suspension of international flights, movement restriction, and the closure of international borders [4]. Public service announcements emphasized policies such as physical distancing, the ban on public gatherings, fewer passengers in public vehicles and frequent handwashing with soap and water.

Hand washing with soap and water is one of the most important public health preventative measures against transmission of the COVID-19 virus [5]. Studies show that access to a convenient handwashing station (HWS) is associated with higher rates of hand washing with soap and water [6–10].

The World Health Organization recommends that handwashing facilities be available at the entrance of all public buildings and transport hubs [11]. The COVID-19 pandemic has affirmed the key role that HWSs equipped with soap and water must play in facilitating handwashing and has sparked community-led and institutional efforts, resulting in significant scale-up of HWSs, especially in public places [12, 13]. Handwashing stations were fronted as one of the critical preventive actions in the first phase of the COVID-19 pandemic, aimed at disrupting virus transmission.

Since the onset of the COVID-19 pandemic, governments, donors and philanthropists made efforts to distribute and install HWSs in public spaces [14–16]. The National Business Compact Coalition (NBCC) in Kenya, formed in March 2020, brought together a network of private sector actors including, Kenya Association of Manufacturers, Kenya Private Sector Alliance, Public Relations Society of Kenya, Amref Health Africa, SDG Partnership Platform and the UN family in Kenya to complement government efforts towards accelerating COVID-19 preventative action [17]. The efforts involved procurement and deployment of public handwashing stations, stockpiling of HWS with soap, distribution of IEC materials, sensitization of HWS caretakers and users on operation and maintenance, and monitoring.

Nevertheless, although their widespread coverage is admirable, empirical evidence on the efficacy of the stations in promoting proper hygiene habits remains scarce. The deficiency of concerted systematic efforts has resulted in insufficient data and information on the various aspects related to the maintenance/management, functionality, usage patterns and long-term sustainability of handwashing stations. The outcome of this is that the impact of the public health intervention of investing in public handwashing infrastructure to battle COVID-19 is not sufficiently understood.

In addition, there is a greater need for in-depth research into the underlying factors that determine the proper preservation and functioning of these infrastructures. Gaining insights into these determinants will be of essence in optimizing handwashing compliance which is imperative for the prevention of the spread of infectious diseases including COVID-19.

The research conducted so far has focused on handwashing behaviour within controlled environments such as healthcare facilities, schools, and homes. However, it is clear that the implementation of monitoring and implementation practices in public spaces is poorly understood [18]. It is this critical gap that has emphasized the necessity for thorough studies aimed at observing and assessing the efficacy of hand washing stations in public places. This study, centred on the accessibility, usability and functionality of appropriate stations, is meant to fill this gap and to help to uncover the vital role they play in disease prevention and control process.

The lived experiences, success, and challenges associated with the operation and maintenance of deployed handwashing stations were investigated in our research. Through the findings of this research, we aim to formulate tangible recommendations in the areas that could improve the efficacy and sustainability of existing hand washing facilities in public areas. In addition, these results will be useful for creating intervention strategies targeting communities in order to maximize the effectiveness of hand hygiene, which leads to improved public health outcomes.

## Materials and methods

### Study design

This was a cross-sectional observational study, with a mixed-methods approach, to assess the accessibility, usability, and functionality of the public HWS deployed by NBCC in 2020, in response to the COVID-19 pandemic. This study design was employed due to its appropriateness to provide insights on the operation and management of public HWS in Kenya.

### Study setting and population

The study was conducted in five counties (Nairobi, Kwale, Embu, Mombasa and Homabay) where the interventions were carried out by the Hygiene Behaviour Change Coalition project. Nairobi is the capital of Kenya and is located in the Central region, Mombasa and Kwale are located in the South Eastern region and Embu and Homabay in the in the Eastern and Western regions respectively (Fig 1). The five counties were sampled through simple random sampling using Microsoft (MS) Excel-generated random numbers. The study targeted handwashing stations deployed by NBCC partners and individuals responsible for the day-to-day operations of the handwashing stations as well as the community benefiting from them.

As part of the intervention, between April and August 2020, NBCC distributed 5,311 HWS to 46 counties in Kenya through its partners; Copia, Rotary, Shujaaz, Sanergy, Shining Hope for Communities (SHOFCO), BRCK and Jonathan Jackson Foundation [17]. Three types of HWS were distributed in three phases. In April 2020 (Phase I), 1500 20-litre-capacity HWS with stool stands were distributed. The cost for one unit was USD 5.76. Between May and June 2020 (Phase II), 2253 60-litre-capacity HWS with fabricated stands, waste-water collection basins and a soap holder were distributed alongside IEC posters. The cost for a unit of this size was USD 30.85. Lastly, lessons from the distribution in Phases I and II informed the distribution in Phase III, which took place between the end of July and August 2020. In this final phase, 1558 brightly coloured 100 litre-capacity HWS with wastewater drainage pipes, fabricated stands, soap and soap holders, and IEC posters were distributed. The cost for one unit was USD 39.60 (Fig 2).

Each handwashing station (HWS) was assigned a caretaker responsible for its operation and maintenance. In the context of this study, operation and maintenance encompassed a range of responsibilities assigned to the handwashing station caretakers. These tasks included ensuring the availability of soap and water, regular cleaning of the handwashing stations, refilling the water containers and restocking soap as needed. Regarding the provision of soap and water, guidelines were established during deployment emphasizing the responsibility of HWS recipients to regularly refill water and replenish the soap once the initial supply of 800mg provided by coalition partners was depleted. This task was undertaken either by the caretakers or other community members. The caretakers fell into two categories: volunteers, typically local business owners or paid individuals. Paid caretakers, predominantly stationed in informal settlements, were contracted by non-governmental organizations affiliated with the National Business Compact Coalition (NBCC). The engagements periods were ranging from 3 to 6

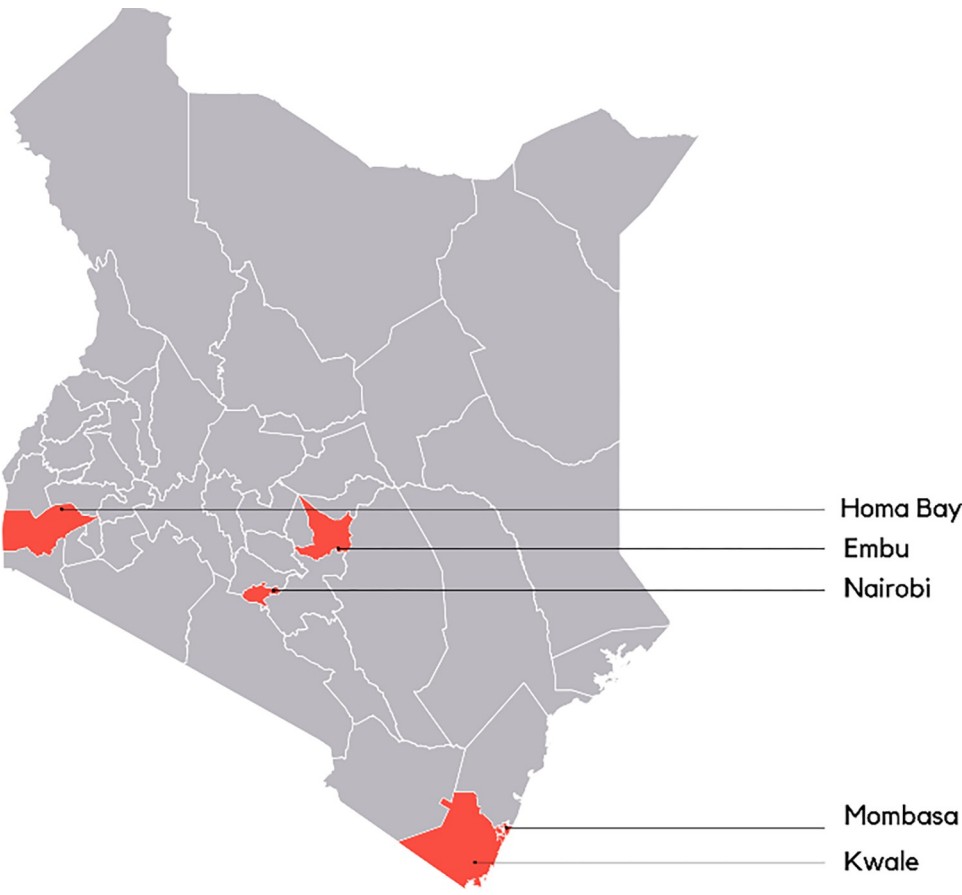

**Fig 1. Map of Kenya with study sites.**

months, coinciding with the peak distribution of HWS and COVID-19 caseload. Their role was to remain on standby, positioned next to or within proximity to the HWS, to ensure optimal operation and maintenance. On the other hand, unpaid caretakers were volunteers residing near the HWS, often near their businesses such as stalls, shops, supermarkets, markets, or transport hubs. These caretakers, presumed to be respected members of the community, were

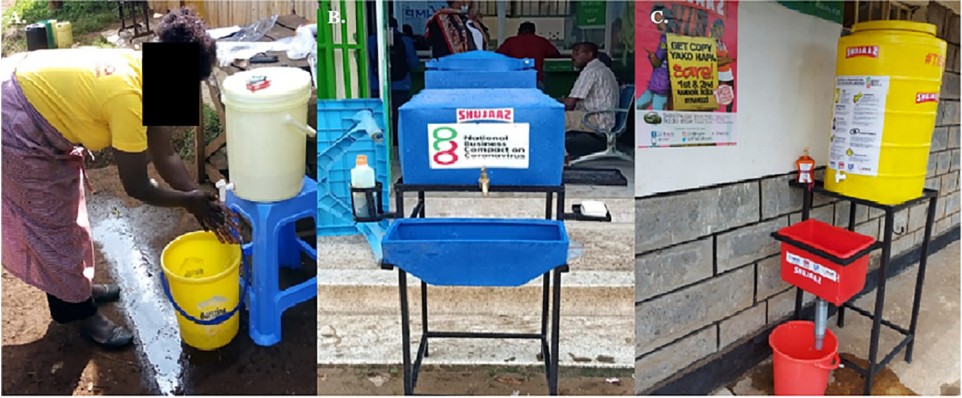

**Fig 2.** Handwashing Stations deployed in phase I, II and III as depicted by panels A, B and C respectively.

expected to assume responsibility for the operation and maintenance of the HWS without formal compensation.

## Study size and sampling procedures

The minimum sample size of 430 HWS was determined using the formula for the finite population from the total distributed 5,311 HWS, while adjusting for a 10% loss of information. The HWSs were randomly selected using a sampling interval, K (5) until the total sample size of 430 HWS was attained. Detailed information regarding the sampling calculation is provided in S1 Equation.

Selected HWSs were located at bus terminals, shopping centres, markets, roadsides, public toilets, and government offices across urban and rural areas of Kenya. S1 Table shows the characteristics of the study sample.

## Data collection methods

Data was collected through qualitative and quantitative techniques for the period between 23rd November 2020 and 20th February 2021. Quantitative methods included spot checks, caretaker surveys, user observation and user exit surveys. Handwashing observation can be subject to reactivity bias and the bias was addressed by passively observing the handwashing facilities and concealing data collection tablets. Qualitative methods included key informant interviews, in-depth interviews and focus group discussions. Spot checks were conducted at all selected HWS using a structured checklist with pre-selected themes which included: location/placement, visibility and accessibility of the HWS and factors relating to the functionality of the facility such as, the presence of running water, availability of soap and functionality of the tap, that is, if it is visible and operational. The caretaker survey was conducted at all HWSs that had a caretaker present to evaluate the operation and management of the facilities. The survey encompassed routine tasks such as repairing broken taps or refilling soap and water to ensure continued functionality. Additionally, the survey evaluated the efficacy of the HWSs by scrutinizing broader factors like community engagement, institutional support and resource allocation. These aspects aim to ensure the long-term viability and effectiveness of the handwashing intervention beyond the initial distribution phase. Furthermore, 2–5 users at each facility were observed by the enumerators using a structured observation tool that recorded gender, approximate age, and handwashing technique used by the users. At each HWS, 2–5 users were purposively sampled for diversity (female, male, older adults and people with visible disabilities) and recruited after using the HWSs. These users were interviewed using a structured survey tool to evaluate their experiences and perceptions of the HWSs.

The study conducted Key Informant Interviews (KIIs) with the technical officers and leadership from the NBCC to provide information on the distribution and operation of the HWSs while Focus Group Discussions (FGDs) were conducted with selected HWS users using an FGD guide, to assess the acceptability, usability and functionality of the HWS; and In-depth Interviews (IDIs) with select caretakers from each sampled county were carried out through mobile phones to provide additional information regarding the operation, management and sustainability of the HWSs. The qualitative interviews were audio-recorded, transcribed verbatim, and translated into English where necessary. Relevant documents related to the cost of implementation of the units were collected and reviewed.

## Data collection and management

The survey was conducted electronically using a mobile phone application–the Open Data Kit (ODK) standard survey tool by trained enumerators who were under the supervision of field

supervisors. The collected Data was transmitted onsite to Amref Health Africa's servers for routine data cleaning and analysis. The Data was de-identified and given unique codes to prevent the identification of individual participants during and after data collection.

## Statistical methods

**Variables.** The outcome variable of interest in this study was the effective use of handwashing stations (HWSs). Effective use was defined based on several criteria observed through caretaker surveys, user observations and user exit surveys. It encompassed factors such as the presence of running water, availability of soap and the functionality of the tap. The independent variables examined in the study included: Behavioural Change Communication, evaluated through responses obtained from caretaker surveys. They included the extent and effectiveness of communication efforts aimed at promoting hand hygiene practices; the level of knowledge and awareness of the accessibility of HWSs, assessed through questions in the caretaker and user exit surveys regarding users' understanding of the purpose and location of the handwashing stations. They additionally included ease of access and availability of handwashing products determined by observing the presence of soap and water at the handwashing stations during user observations.

## Quantitative data

Statistical analysis was conducted using SPSS (Version 20). Descriptive characteristics were presented as frequency and percentages. Chi-square tests or where applicable Fisher's exact tests were used to compare characteristics of HWSs between paid and unpaid caretakers; and were also applied to determine HWSs characteristics associated with the observed number of individuals within each HWSs. The handwashing stations (HWSs) investigated in this study were assessed based on several key characteristics. First off, the availability of handwashing commodities such as soap, water and functioning taps was examined to ensure the stations were equipped for effective hand hygiene practices. Additionally, the location of the HWSs within the community or public spaces was considered, ranging from markets to transportation hubs, influencing their accessibility and visibility to users. Accessibility was evaluated in terms of ease of use, including factors like the presence of ramps for individuals with disabilities or proximity to high-traffic areas. Moreover, the visibility of the stations, ensuring they were easily noticeable to users, was also an important aspect assessed. Finally, maintenance of the HWSs, encompassing routine upkeep tasks, like repairing broken taps and refilling of soap and water was evaluated to ensure ongoing functionality and hygiene standards. Assessing these characteristics provided valuable insights into the effectiveness and usability of the handwashing stations in promoting hand hygiene practices within the community or public settings. For the analysis assessing the association between the presence of information, education, and communication (IEC) materials and handwashing technique, each handwashing observation was treated as an individual data point. The unit of analysis was defined as each instance of handwashing was observed at a handwashing station (HWS). Multi-nominal logistic regression was used to determine predictors (HWSs location, visibility, accessibility, handwashing commodities and operation and management of the observed number of individuals within each HWS. The only predictors that were significantly associated with the outcome in the chi-square test of association were included in a multinomial regression model.

## Qualitative data

The translated transcripts were exported from MS Word to NVivo 11 software. All the transcripts were read, and the codes were deductively developed and defined through discussion

and reflection. The codes were then merged into the pre-identified themes that were in line with the objectives and applied to all the transcripts. These results were used to support the findings from the quantitative results.

## Ethics approval and consent of participation

Ethical Approval for this study was sought from Amref's Ethical and Scientific Review Committee (ESRC) in Kenya (Reference number: P875_2020) and the London School of Hygiene & Tropical Medicine (UK) (reference number: 22704). The principal investigator and the co-investigators ensured that the study adhered to all the standard ethical practices with all the investigators completing an ethical certification course. To gain the participants' informed consent, the research aims and processes were first explained to all participants and thereafter, those who could read and write were asked to sign physical forms, while those who could not read and write were asked for a thumbprint. A literate witness (such as a family member, person from a neighboring house or passerby) were asked to sign to validate the research participant's thumbprint. In addition, verbal consent was obtained for the telephone interviews and the face-to-face interviews with the participant's information sheets read out to participants. Finally, relevant permissions were sought from the respective county health management teams and authorities before data collection.

## Results

A total of 430 HWS were targeted for this study conducted between November and December 2020, 6–8 months after the distribution of the HWSs. Of these, 316 (73%) were located and surveyed. Of the ones not located (114), 58 (51%) could not be traced because caretakers were absent, 12 (10.5%) were those where the caretakers had relocated their business, 15 (14%) were at the caretakers' homes, 5 (4.4%) had claims from caretakers that movement of the HWSs was tedious, 4 (3.5%) had been stolen and 16 (14%) were in storage and therefore not in use (Fig 3). In addition, 12 (4%) of the located HWSs did not have a caretaker to manage them. These results represent the HWSs that were located.

Overall, a total of 578 user exit interviews, 411 user observations, five Focus Group Discussions, nine in-depth interviews, and seven key informant interviews were conducted. From the direct observations, the handwashing stations were used by people across different age groups, with 88.6% (365) adults, aged between ages 18 and 59 years, observed washing their hands, while preschool children and elderly users were the least users at 1% (4) and 1.9% (8) respectively.

The HWSs were present in shopping centres (173, 55%), markets (34, 11%), bus terminus (19, 6%), government facilities (16, 5%), places of worship (8, 3%), and community halls (5, 2%). Others were present in health facilities (4, 1%), next to public toilets (9, 3%) and schools (8, 3%).

## Accessibility of handwashing stations in public spaces

Of the 316 located HWSs, 193 (61%) were easily visible within the vicinity of its setting, that is, from anywhere in the market or bus terminal. In contrast, 117 (37%) HWSs could be spotted only if a user knew their exact location (S2 Table). The study findings also established that 51% of the HWSs had IEC materials (posters) to promote proper handwashing techniques. Of these, 45% were at eye level and easy to read while using the facility (Fig 4).

The study findings indicate that 246 (78%) of the located HWSs could be conveniently reached by everyone in the vicinity, while the other 22% could only be reached by those near them. That is, some people could see the HWSs but could not reach them because of physical

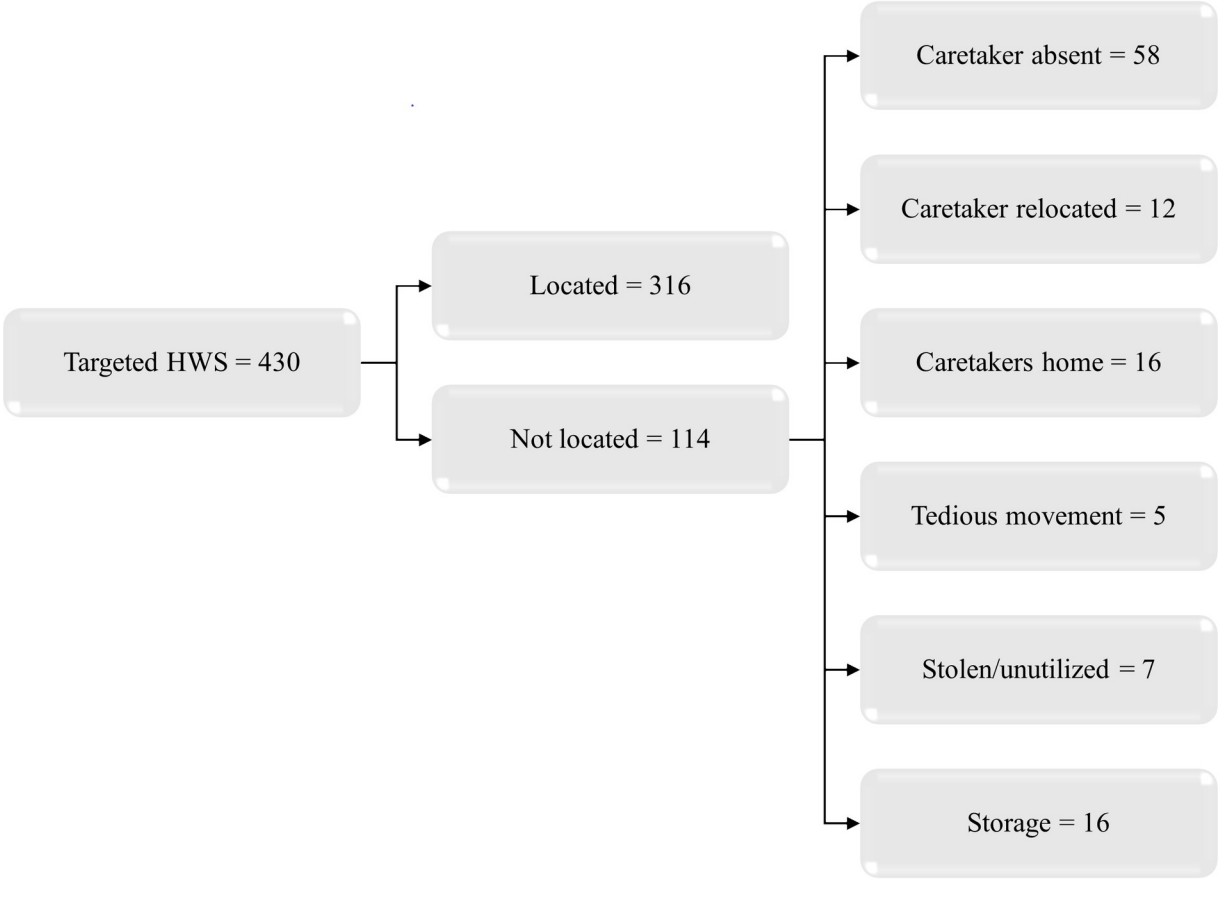

**Fig 3. Study flow diagram.**

barriers like busy roads or a tall fence around a market. The findings indicated that 58% of male and 31% of female adults easily reached and used the HWSs with no notable challenges. Moreover, 35%, 77%, and 77% of the HWSs were reported by the enumerators to be accessible to persons using wheelchairs, children under the age of 12 years, and the elderly, respectively (S2 Table).

During the focus group discussions, one participant highlighted an accessibility issue, stating that; ". . .at the roundabout, the HWS is not accessible to people with disability. It is raised and cannot be accessed by someone using a wheelchair." (FGD-User- Respondent 1 in Homabay).

The presence of the HWSs in public spaces was reinforced by the Kenya's Ministry of Health through a national campaign for mandatory handwashing in public spaces, The Ministry also heavily invested in a number of communication campaigns that led to mass awareness and mobilization for regular handwashing. Qualitative findings from in-depth interviews affirmed the same;

". . .advertisements and messaging were all over on proper handwashing . . ." (IDI-Caretaker-Mombasa).

There was a significant difference in handwashing techniques between observations made at handwashing stations (HWSs) with IEC materials compared to those without (41% vs 58%,

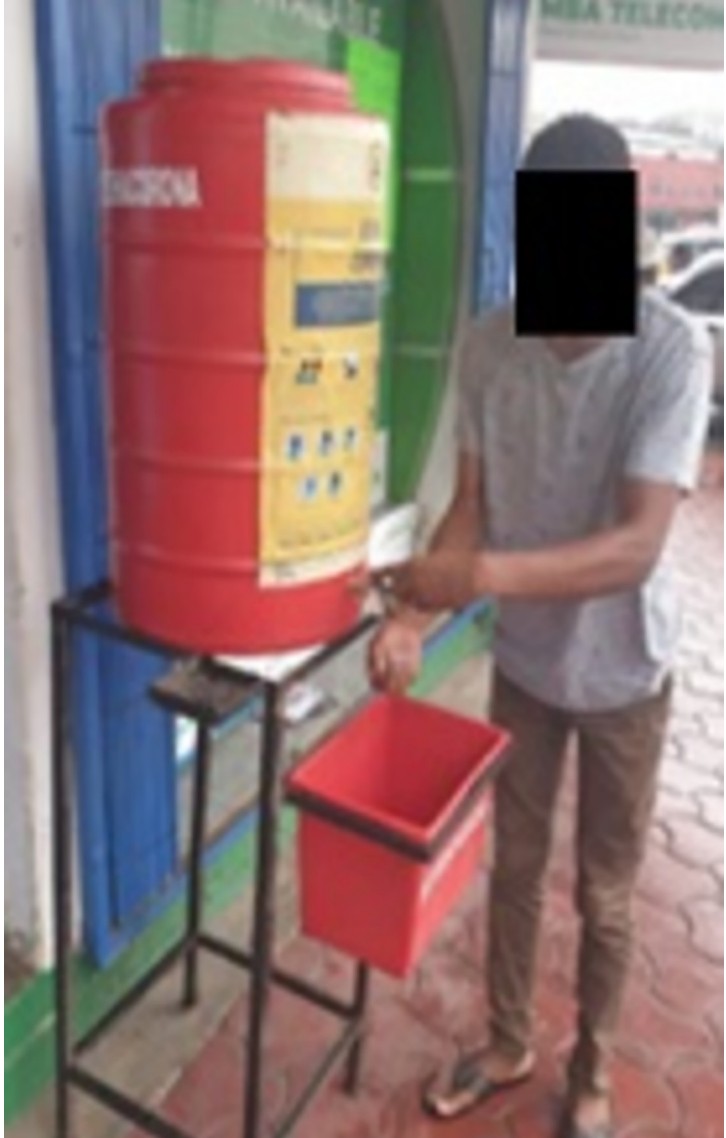

**Fig 4. Pictures demonstrating IEC materials placed at eye level and easy to read while using the facility.**

p<0.001). Each handwashing observation was treated as an individual data point for analysis (S3 Table).

## Usability of the public HWS

Lack of drainage for wastewater was identified as a barrier to use in 20% of the HWSs, with users reporting being put off by the water splashing on their feet while washing their hands. In addition, from observations, users in 8% of HWSs had to significantly bend to wash their hands, and about half (49.1%) of the HWSs were set up on uneven and/or muddy ground (S4 Table).

The qualitative findings affirm these findings with respondents reporting:

*". . .the challenge users experience when washing hands is water splashes on them." (IDI-Caretaker in Nairobi)*

Half of the users (51%) reported that their physical reach and use of HWSs, was influenced by the observed neatness of the sites around the HWSs, or the 'cleanliness' of the bar soap, because they feared contamination and subsequent risk of COVID-19 (S4 Table). These sentiments were also expressed by FGD participants who highlighted experiences of perceived cleanliness of the HWSs environment and soap.

*"There are places the soap is too dirty and even turning black, yet it was a white bar soap. It is touched by many people and nobody bothers to wash it, so when you see it, you don't even want to touch it. . . I have never washed my hands with dirty soap. I'd rather wash with water only. So many people touch the soap."* (FGD MSAR4)

*"if the bucket or the soap is dirty, I won't use it." (FGD-User- Respondent 2-Kwale)*

## Functionality of the handwashing stations in public spaces

Of the 316 HWSs observed by the enumerators, 76% had sufficient water levels (more than half the tank), 17% had some water in the tank and 7% had no water in it. Soap was available at 89% of the HWSs. The type of soap present varied from liquid (62%) to bar (26%) with a small percentage (2%) having soapy water. Taps were functioning at 96% of the HWSs, but 6% of these were leaking. Qualitative findings pointed to the high cost of consumables and scarcity of water in other areas as challenges that affect the functionality of HWSs (Table 1).

## Operation and maintenance of public HWS

Caretakers in 304 (96%) of the located HWSs participated in the study. Of those, 209 were unpaid caretakers and 95 were paid caretakers. Half of the caretakers (52%) reported having received training on the maintenance of the HWSs. The training was more readily accepted by paid caretakers than by unpaid caretakers (71.6% vs 43.1%, *P<0.001)* (Table 1). Most of the caretakers (80%) reported visiting the HWSs multiple times a day to check on water levels, soap, cleanliness, whether the tap was working, and if the wastewater basin was full (Table 1).

One of three caretakers reported on water scarcity, as a regular reason for the HWSs being out of water. Half of the caretakers did not have a system to notify them when their HWSs were out of the water (Table 1). The study established that the majority of caretakers (84%), reported replenishing water by themselves, with a fraction (16%) doing it jointly with their neighbors. Some potential setbacks identified from the caretaker survey findings suggest that the caretakers reported cumbersomeness (50%) in refilling the tanks, with 28% of them reporting being unable to refill water in one or more instances. The reasons given by the caretakers during the FGDs included the lack of a steady water supply in the area and their inability to buy and afford water.

*"Water is very expensive. . .. we buy water from vendors and they charge a lot of money." (KII-partner organization working in informal settlements).*

Most HWSs with a paid caretaker had liquid soap available (96%), whereas HWSs with unpaid caretakers had a variety of soap types available: liquid soap (46%), bar soap (36%) and soapy water (2%). Enumerators noted that HWSs with unpaid caretakers lacked soap more often than the HWSs managed by a paid caretaker (15% vs 2%).

**Table 1. Comparison of functionality and management factors of handwashing stations managed by paid vs unpaid caretakers.**

| | All HWS, n = 316 | HWS with unpaid caretaker, n = 218 | HWS with paid caretaker, n = 98 | p-value |
|---|---|---|---|---|
| **HWS is functional, n (%) *** | 265 (83.9%) | 171 (78.4%) | 94 (95.9%) | <0.001 |
| **Presence of water in the HWS** | | | | |
| **No water** | 22 (7.0%) | 19 (8.7%) | 3 (3.1%) | <0.001 |
| **There is very little water in the tank** | 54 (17.1%) | 47 (21.6%) | 7 (7.1%) | |
| **The water level is sufficient/satisfactory** | 240 (75.9%) | 152 (67.7%) | 88 (89.8%) | |
| **Person in charge of replenishing water, n (%)** | | | | |
| **Caretaker** | 256(84.2%) | 174(83.3%) | 82(86.3%) | 0.497 |
| **Other individuals e.g., neighbours** | 48(15.8%) | 35(16.7%) | 13(13.7%) | |
| **Difficulty sourcing water to refill the tank, n (%)** | | | | |
| **Yes** | 136(44.7%) | 87(41.6%) | 49(51.6%) | 0.106 |
| **No** | 168(55.3%) | 122(58.4%) | 46(48.4%) | |
| **Soap available for use** | | | | |
| **Bar soap is available** | 81 (25.6%) | 79 (36.2%) | 2 (2.0%) | <0.001 |
| **Liquid soap is available** | 195 (61.7%) | 101 (46.3%) | 94 (96.0%) | |
| **Soapy water is available** | 5 (1.6%) | 5 (2.3%) | 0 (0.0%) | |
| **Soap is not available** | 35 (11.1%) | 33 (15.1%) | 2 (2.0%) | |
| **Person in charge of replenishing soap, n (%)** | | | | |
| **Caretaker** | 271 (89.1%) | 180 (86.1%) | 91 (95.8%) | 0.012 |
| **Other individuals** | 33 (10.9%) | 29 (13.9%) | 4 (4.2%) | |
| **Location of the soap** | | | | |
| **Soap available around the HWSs vicinity** | 276 (98.2%) | 180 (97.3%) | 96 (100%) | 0.011 |
| **In physical custody of the caretaker (inside the business premise)** | 5 (1.8%) | 5 (2.7%) | 0 | |
| **Timing of soap replacement, n (%)** | | | | |
| **When it reaches a low level/pre-set level** | 244 (80.3%) | 161 (77.0%) | 83 (87.4%) | 0.036 |
| **When there is no soap at the station** | 50 (16.4%) | 42 (20.1%) | 8 (8.4%) | |
| **Other** | 10 (3.3%) | 6 (2.9%) | 4 (4.2%) | |
| **Breakages/cracks visible on HWS** | 21 (6.6%) | 12 (5.5%) | 9(9.2%) | 0.225 |
| **Status of Tap** | | | | |
| **Faulty–not working** | 8 (2.5%) | 7 (3.2%) | 1 (1.0%) | 0.033 |
| **Working but leaking** | 17 (5.4%) | 16 (7.3%) | 1 (1.0%) | |
| **Working without leaking** | 291 (92.1%) | 195 (89.4%) | 96 (98.0%) | |
| **Caretaker was located, n (%)** | 304 (96.2%) | 209 (95.9%) | 95 (96.9%) | <0.646 |
| **Received training on HWSs maintenance, n (%)** | 158 (52.0%) | 90 (43.1%) | 68 (71.6%) | <0.001 |
| **How often does the caretaker visit the HWS, n (%)** | | | | |
| **Multiple times a day** | 242 (79.6%) | 160 (76.6%) | 82 (86.3%) | 0.189 |
| **At least once a day** | 50 (16.4%) | 41 (19.6%) | 9 (9.5%) | |
| **At least twice a week** | 10 (3.3%) | 6 (2.9%) | 4 (4.2%) | |
| **Less than once a week** | 1 (0.3%) | 1 (0.5%) | 0 | |
| **Whenever I am notified that there is a problem** | 1 (0.3%) | 1 (0.5%) | 0 | |
| **Unable to refill water because it was unavailable** | | | | |
| **Yes** | 85(28.0%) | 48(23.0%) | 37(38.9%) | 0.004 |
| **No** | 219(72.0%) | 161(77.0%) | 58(61.1%) | |
| **HWS is cleaned by, n (%)** | | | | |
| **Caretaker** | 255 (83.9%) | 168 (80.4%) | 87 (91.6%) | 0.045 |
| **Other individuals** | 48 (15.8%) | 40 (19.1%) | 8 (8.4%) | |
| **No one** | 1 (0.3%) | 1 (0.5%) | 0 | |

*(Continued)*

**Table 1.** (*Continued*)

|  | All HWS, n = 316 | HWS with unpaid caretaker, n = 218 | HWS with paid caretaker, n = 98 | p-value |
|---|---|---|---|---|
| **Washed hands using proper technique** * (n = 411) | 203 (49.4%) | 99 (39.8%) | 104 (64.2%) | <0.001 |

\* Functional handwashing facility was defined as having the following present at the time of observation: water in the tank, soap located on or next to the facility, and a tap that is working.

\*\* Proper technique as defined by WHO is the act of cleaning one's hands with soap and running water for at least 20 seconds (20).

*"The problem comes in with soap. . .you find that soap is usually not available, and when it is, it does not last because there are many people. . .Even if you buy it yourself, it won't last. By 11 am, it is usually depleted." (FGD-caretakers in Embu-Respondent 9).*

Some caretakers feared the loss of business if they were too strict in enforcing the government directives, that required customers to properly wash their hands before being served.

*". . .customers are different. If you instruct some of them on how to properly wash their hands, you risk losing the customer." (IDI-Caretaker-Embu)*

HWSs were cleaned regularly by 80.4% of unpaid caretakers, and 91.6% of paid caretakers (p = 0.045). The majority of HWSs were managed by an individual caretaker while others were managed collectively by neighboring shops/community members. The qualitative findings affirm these:

*". . .even when it breaks down, I am the one who repairs it. When water is depleted, I am the one. . . When soap is depleted, I am the one. . ." (IDI-Caretaker- Mombasa)*

*"I received support from my neighbours to refill water. It is not very complicated. . .in the evening, I always ask who will wake up in the morning and place it outside. . .." (IDI-Caretaker-Kwale)*

*". . .for soap (replenishment) we contribute sometimes if it is there it is a shared responsibility. When one sees it getting depleted, they (neighbouring shops and other community members) replace it." (IDI-Caretaker-Kwale)*

However, some of the HWS users reported being put off by negative attitudes by the caretakers, and the perceived undue pressure of buying from some shops beyond using the HWSs.

*". . .the attitude of the owner puts you off when they tell you that it is not a public facility and you should go wash somewhere else." (FGD-User- Respondent 3-Homabay)*

*". . .some shops do not allow you to wash your hands before you buy. After you buy that is when they give you the permission to wash your hands." (FGD-User- Respondent 9-Homabay).*

There was a significant difference between the number of users that washed hands using the proper technique at a HWS with a paid caretaker, versus HWSs with an unpaid caretaker (64% vs. 40%, p<0.001).

## Discussion

Our study of public HWSs carried out in response to the COVID-19 pandemic in Kenya, found that one out of four HWSs distributed, went missing within six to eight months after distribution. This loss rate is concerning and raises serious questions about the sustainability of the distribution of HWSs in public places. As our sample is representative of the distribution of 5,311 HWSs across Kenya, this loss rate represents a potential loss of USD37,076 in HWSs purchase costs, not including logistics and training. Global technical guidance for the distribution of facilities, often lists theft as a concern when distributing public HWSs [19, 20]. However, our study found that the most common reasons for loss are due to the caretakers moving the HWSs. This leads to HWSs not being utilized or being used in a different location and not the hotspot identified by the distributor. After the distribution of the HWSs, caretakers received support from NBCC in the form of trainings, distribution of IEC materials and soap, procured through a fund that was established to respond to the pandemic. An improved system for follow-up could help increase accountability by caretakers, providing an incentive for them to ensure the continued use of the HWS. Another measure that may mitigate this issue is the use of facilities that are securely fixed or permanent, piped facilities [19].

A motivated caretaker was the greatest enabler for the proper management of the HWSs. The study found that paid caretakers were responsible for a higher proportion of functional handwashing facilities compared to unpaid caretakers. Furthermore, HWSs managed by paid caretakers were more likely to stock liquid soap as it was preferred by users over bar soap and acted as an incentive for members of the public to use the HWSs. Facilities managed by paid caretakers were more likely to have satisfactory levels of water (90% vs. 68% p<0.001). HWSs with paid caretakers had significantly more users washing their hands using the proper technique. This is likely due to the presence of the paid caretaker guiding users to wash their hands properly [21, 22].

A larger proportion (71.6%) of paid caretakers were trained on the maintenance of HWSs compared to unpaid caretakers (43.1%). However, there was no significant difference in the state of the HWSs (e.g. cracks or breakages visible on the HWSs). There was also no notable difference in the functioning of taps. It is critical to train the HWS caretaker on appropriate operation and management, including positioning, securing, cleaning and disinfection of touch points, regular refilling of handwashing commodities, repair needs, and sourcing and fixing of replaceable parts. Operation and management plans should also be developed where appropriate [23]. Regular operation and maintenance arrangements for handwashing facilities in public places, including clear responsibilities and budgets will contribute to the sustainable use of the HWSs [20].

Although HWSs managed by paid caretakers were overall more functional and better managed, it should be noted that more than 75% of HWSs managed by unpaid caretakers were functional, meaning water was present in the tank, soap located on or next to the facility and the HWS had a working tap. Of the HWSs sampled for this study, 66% were maintained by staff and business owners at markets and shopping centres. Interviews with users revealed that pressure to purchase items from the retailers after the use of the HWSs could act as a barrier to using the facility. Unpaid caretakers were proud to maintain the units and to offer the public an option for hand cleansing in public, but found the lack of financial support challenging when the need to replenish consumables such as soap and water arose.

This study found that the presence of IEC materials on the HWSs, and the presence of a paid caretaker by the HWSs, were positively associated with users washing their hands with soap and water for at least 40 seconds. Whereas the latter can be costly, the addition of printed IEC materials on the units was identified as a low-cost intervention that should be implemented along with future public HWS distributions.

Only 35% of HWSs surveyed were accessible for wheelchair users. This is a clear indication that the distribution of HWSs by the NBCC failed in locating suitable placements for the HWSs where they could be accessible to all. Recommendations by implementing agencies suggest placement of the HWSs on flat and clear surfaces, that enable the facility to be easily accessed by wheelchair users or those with mobility challenges [19, 24]. If necessary, adding ramps made of suitable and non-slippery materials could be used to make non-accessible locations accessible. Another inclusive option is HWSs customised for wheelchair users, as utilised by WaterAid in Zambia [19].

Additionally, this study points out that design considerations on the drainage of wastewater, height, and soap placement affect the usability of HWSs. Other studies support this finding as they suggest that the design features provide a pleasant and convenient user experience for all users [25, 26]. Users expressed concern about bar soaps and HWSs being perceived as dirty and a source of potential transmission of pathogens, due to the number of people that touch the soap at the HWS before them. Providing liquid soap and clean facilities may increase the number of people washing their hands and washing using proper techniques [24, 25, 27]. Furthermore, the study findings point to the need for user-friendly, hands-free mechanisms such as foot pedals, to further mitigate the issue of re-contamination of hands after using the HWSs [19, 28]. Interviews with users revealed that the use of larger wastewater basins to avoid splashing, would have improved the user experience. This would have avoided the ground becoming a slipping hazard, a finding corroborated by other implementing actors [19].

This study has some limitations. First, self-reported data from the caretakers on the management of the handwashing stations, could have been influenced by social desirability bias [29]. Second, data on the frequency of use of the HWSs was not collected. Monitoring handwashing behavior in public settings is notoriously difficult [18]. The HWSs were placed in a variety of locations such as markets, bus stations and places of worship. These locations have different opening hours, some are open day and night, and some are not open all days of the week. Not all of them have clear points of entry and exit. This made it challenging to measure the frequency of use, due to the lack of an accurate denominator. Providing functional HWSs that are accessible and usable, is not sufficient to ensure increased uptake and correct use of the HWSs. A study in the Democratic Republic of Congo, found that only 10% of people utilized available hand hygiene facilities, before entering public buildings during the COVID-19 pandemic [30]. Future studies should utilize recommendations by the Hand Hygiene for All initiative, to guide monitoring hand hygiene in public spaces [18].

The findings of this study align with the principles of the Technology Acceptance Model (TAM), shedding light on the functionality, usability, and acceptability of handwashing stations in public settings. Our observations revealed high usage rates among adults, indicating the perceived usefulness of these facilities in promoting hand hygiene practices. The presence of handwashing stations across various public spaces underscores their accessibility, with factors such as visibility and proximity influencing user behavior, consistent with TAM's construct of perceived ease of use.

Furthermore, our study identified barriers to usability, such as inadequate drainage and physical barriers, which may deter users from utilizing handwashing stations effectively. These findings emphasize the importance of addressing usability concerns, to enhance technology adoption and utilization, in line with TAM's focus on external factors influencing perceived ease of use.

Regarding behavioral intentions and usage behavior, this study found a positive correlation between users' intentions to use handwashing stations and their actual usage behavior. This aligns with TAM's assertion that behavioral intentions are strong predictors of technology adoption. However, challenges related to operation and maintenance, such as water scarcity

and caretaker training, may hinder users' ability to access and utilize handwashing stations effectively. This highlights the importance of addressing operational barriers to enhance technology acceptance and sustainability.

## Conclusion

The provision of public handwashing facilities in response to the COVID-19 pandemic was a widely utilized intervention. This study has demonstrated that without long-term investment from the implementing agency, the distribution of public handwashing facilities will have short-lived impact. It was identified that only 61% of handwashing facilities targeted in this study were optimally located and functional. The missing HWSs were majorly due to their relocation by caretakers moving the HWSs, and not theft. This study also found that recommendations for making HWSs accessible for wheelchair users and those with mobility challenges were not adhered to when the facilities were being placed. Future placement plans should consider accessibility measures for wheelchair users and those with mobility issues when planning distributions, as well as plans for long-term support for caretakers. The study did not measure the impact of public handwashing facilities on COVID-19 control. Future research should aim to evaluate the actual frequency of use of the HWSs.

## Supporting information

**S1 Equation. Sampling calculation for determining study size.**
(DOCX)

**S1 Table. Sample distribution by county, partner, type of caretaker and the phases of distribution.**
(DOCX)

**S2 Table. Accessibility of handwashing stations in public spaces.**
(DOCX)

**S3 Table. Association between washing hands using proper technique and the presence of information, education and communication materials on the hand washing station.**
(DOCX)

**S4 Table. Usability of located HWS deployed for COVID-19 response in Kenya.**
(DOCX)

## Acknowledgments

We are grateful to all the HWS caretakers and respondents who took time to respond to the survey and provide other information about the implementation of the public handwashing station interventions in Kenya. The team is also grateful to the leadership at Amref Health Africa in Kenya and the entire secretariat at the National Business Compact on Coronavirus in particular Dr Myriam Sidibe and Maggie Rarieya, for their support through the entire process. Thanks also to Robert Dreibelbis for reviewing this manuscript.

## Author Contributions

**Conceptualization:** Josphat Martin Muchangi, Dennis Munai, Rogers Moraro, Viola Tupeyia, Jennifer Lamb, Richard Gichuki, Katie Greenland.

**Data curation:** Dennis Munai, Rogers Moraro, Jennifer Lamb, Richard Gichuki, Sheillah Simiyu.

**Formal analysis:** Viola Tupeyia, Richard Gichuki.

**Investigation:** Josphat Martin Muchangi, Dennis Munai, Rogers Moraro, Viola Tupeyia, Judy Muriithi, Richard Gichuki.

**Methodology:** Josphat Martin Muchangi, Rogers Moraro, Viola Tupeyia, Richard Gichuki.

**Project administration:** Josphat Martin Muchangi, Dennis Munai, Rogers Moraro, Viola Tupeyia.

**Supervision:** Josphat Martin Muchangi, Dennis Munai, Judy Muriithi.

**Validation:** Josphat Martin Muchangi, Dennis Munai, Katie Greenland, Sheillah Simiyu.

**Visualization:** Dennis Munai, Astrid Hasund Thorseth, Viola Tupeyia, Judy Muriithi, Jennifer Lamb, Sheillah Simiyu.

**Writing – original draft:** Dennis Munai, Rogers Moraro, Astrid Hasund Thorseth, Viola Tupeyia, Judy Muriithi.

**Writing – review & editing:** Josphat Martin Muchangi, Astrid Hasund Thorseth, Jennifer Lamb, Katie Greenland, Sheillah Simiyu.

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
