## [Decision Letter · Decision Letter 0]

26 Dec 2023

PONE-D-23-25273Lessons from the deployment and management of public handwashing stations in response to the COVID-19 pandemic in Kenya: A cross-sectional, observational studyPLOS ONE

Dear Dr. Munai,

Thank you for submitting your manuscript to PLOS ONE. After careful consideration, we feel that it has merit but does not fully meet PLOS ONE’s publication criteria as it currently stands. Therefore, we invite you to submit a revised version of the manuscript that addresses the points raised during the review process.

**ACADEMIC EDITOR: **Please see editor's comments below

We look forward to receiving your revised manuscript.

Kind regards,

D. Daniel, Ph.D.

Academic Editor

PLOS ONE

Journal Requirements:

2. Please report in the Methods section the day, month and year of the start and end of the participant recruitment period for this study. 

5. In this instance it seems there may be acceptable restrictions in place that prevent the public sharing of your minimal data. However, in line with our goal of ensuring long-term data availability to all interested researchers, PLOS’ Data Policy states that authors cannot be the sole named individuals responsible for ensuring data access (http://journals.plos.org/plosone/s/data-availability#loc-acceptable-data-sharing-methods).

6. We note that Figures 1 and 4 in your submission contain copyrighted images. All PLOS content is published under the Creative Commons Attribution License (CC BY 4.0), which means that the manuscript, images, and Supporting Information files will be freely available online, and any third party is permitted to access, download, copy, distribute, and use these materials in any way, even commercially, with proper attribution. For more information, see our copyright guidelines: http://journals.plos.org/plosone/s/licenses-and-copyright.

a. You may seek permission from the original copyright holder of Figures 1 and 4  to publish the content specifically under the CC BY 4.0 license. 

7. We note that Figure 2 in your submission contain [map/satellite] images which may be copyrighted. All PLOS content is published under the Creative Commons Attribution License (CC BY 4.0), which means that the manuscript, images, and Supporting Information files will be freely available online, and any third party is permitted to access, download, copy, distribute, and use these materials in any way, even commercially, with proper attribution. For these reasons, we cannot publish previously copyrighted maps or satellite images created using proprietary data, such as Google software (Google Maps, Street View, and Earth). For more information, see our copyright guidelines: http://journals.plos.org/plosone/s/licenses-and-copyright.

Additional Editor Comments:

Please reduce the words in the abstract to 250 words.

I don’t see the knowledge gap in the introduction

I think lines 81-91 on page 4 should be put in the method, also the line

I suggest removing the sampling calculation to supplementary material

Line 199-202 -> Why is the font size different?

Line 246-249 -> only one asterisk for two footnotes?

Line 259-260 -> It is strange that the quote appears here without any explanation/narration before.

Table 3 -> Please report statistical results properly.

I think the 5 tables (and some of them are big tables) are too much. Please consider reducing them and putting some in the supplementary.

There should be a conclusion section. Please make the last paragraph in the discussion as your conclusion section.

Reviewers' comments:

Reviewer's Responses to Questions

**Comments to the Author**

1. Is the manuscript technically sound, and do the data support the conclusions?

Reviewer #1: Yes

Reviewer #2: Yes

2. Has the statistical analysis been performed appropriately and rigorously? 

Reviewer #1: Yes

Reviewer #2: Yes

3. Have the authors made all data underlying the findings in their manuscript fully available?

Reviewer #1: Yes

Reviewer #2: Yes

4. Is the manuscript presented in an intelligible fashion and written in standard English?

Reviewer #1: Yes

Reviewer #2: Yes

5. Review Comments to the Author

Reviewer #1: Thank you for the opportunity to review the manuscript titled: Lessons from the deployment and management of public handwashing stations in response to the COVID-19 pandemic in Kenya: A cross-sectional, observational study

The authors sought to assess the functionality, usability and accessibility of rolled public handwashing stations in a COVID-19 pandemic context to improve their operation and maintenance and to inform strategies for future rollouts. The authors referenced multiple complementary methodologies and primarily found that the distribution of public handwashing facilities can have a limited lifetime, highlighting some implications of their findings. On the whole the manuscript was an interesting read and of value to PLoS One. There are some opportunities to bolster the manuscript. The main feedback is that additional clarification of the methods/measures and alignment of the results per the listed methodologies is sorely needed.

Here are my detailed comments and suggestions:

Abstract

Line 21: clarify that financial support was also provided

Include other results- qualitative and exit surveys. Quantitative data were collected through spot checks, caretaker surveys, user observations and user-exit interviews; whereas qualitative data were collected using key informant interviews, focus group discussions and in-depth interviews

Introduction

89: Include that fabricated stands, soap and soap holders were also a part of phase III.

98: define was it meant by operation and maintenance (O&M) and the specific tasks the caretakers were to carry out

It is unclear how soap and water were provided

Methods

111: Clarify that it is a mixed method study design

156: mention the factors relating to the functionality of the facility

158: Clarify how maintenance is different from sustainability

173 Quantitative variables is the wrong header, consider data collection and management

183 Define status of soap, water, and tap. Also define HWSs characteristics

(handwashing commodities, location, visibility, and accessibility). How was number of people seen within a HWSs measured?

190: describe the outcome variable of the multinomial regression

198: move to section before qualitative data. How was effective use of HWS, behavioral change communication, level of knowledge and awareness of the accessibility of HWS and availability of handwashing products defined. Clarify variables by data sources- caretaker surveys, user observation, and user exit surveys.

Results

228-229: Confirm the distribution of the overall study population- all the exit users interviewed. The distribution of 97% by study population has limited value. Consider presenting row percentages.

Table 2: Clarify what HWS accessible to children under 12yrs/ older persons means. For example, is this a height requirement?

The footnotes do not have a reference on the table

270-272: It is unclear how this analysis was conducted. The unit of analysis appears to be the HWS but the indicator being measured is handwashing technique. Was only one handwashing experience observed at each HWS? To clarify this discrepancy I recommend making the unit of analysis each handwashing observation (the 411 user observations).

Reviewer #2: Introduction:

The introduction has a clear practical reason of evaluating the intervention. However, some theoretical review perhaps can be added to explain about factors that influence the effective use of technology that guide the development of study instrument and discussion.

Methods:

Very clear

Results:

Figure 3 was very dark

It is not clear which one is the results from multinomial regression model

Is there any analysis to look at the influence of public place and the functionality of the HWS? In some cases, more informal sites will have more risk of not functioning facilities.

Discussion:

It has provide good discussion about the HWS provision intervention and how to increase lifetime and usability. It would be better to integrate the discussion with any model or theory to discuss the importance of all significant factors. Appropriate technology criteria, sustainable technology concept, or technology acceptance model can be some of important concept can be used in discussing the findings.

Conclusion:

Would be better to have a sub title for conclusion

6. PLOS authors have the option to publish the peer review history of their article (what does this mean?). If published, this will include your full peer review and any attached files.

Reviewer #1: **Yes: **Bolanle Olapeju

Reviewer #2: **Yes: **Ni Made Utami Dwipayanti

---

## [Author Response · Author response to Decision Letter 0]

8 Mar 2024

Academic Editor:

Response:

We appreciate the opportunity to address this matter and ensure that our manuscript adheres to PLOS ONE's style guidelines, including those for file naming. Upon review, we have carefully revised the manuscript files to meet the specified style requirements. We have meticulously formatted the document and ensured that all file names comply with the prescribed conventions outlined by PLOS ONE. We apologize for any oversight in our initial submission and appreciate your guidance in ensuring that our manuscript meets the necessary standards.

2. Please report in the Methods section the day, month and year of the start and end of the participant recruitment period for this study.

Response: 

Thank you for your suggestion, the recruitment period for this study has now been included from lines 157 and 158 in the data collection methods.

Response:

Thank you for your suggestion. The manuscript has undergone copyedit. The contacts of the copy editor is as below:

Name: Joyce Karitu

Cell: +254721543021

Email: joycelesham@gmail.com

4. In this instance it seems there may be acceptable restrictions in place that prevent the public sharing of your minimal data. However, in line with our goal of ensuring long-term data availability to all interested researchers, PLOS' Data Policy states that authors cannot be the sole named individuals responsible for ensuring data access (http://journals.plos.org/plosone/s/data-availability#loc-acceptable-data-sharing-methods). 

Response:

We appreciate the opportunity to fulfil this requirement to provide non-author information for a data access committee, ethics committee, or other institutional body to which data requests may be sent in relation to our manuscript. Please find below the contact information for the designated institutional body responsible for handling data access requests:

• Institution Name/Department: Amref Ethics and Scientific Review Committee 

• Contact Person's Name: Mr. Samuel Muhula

• Contact Person's Position: Senior Manager, Learning and Impact

• Contact Person's Email: Samuel.muhula@amref.org

• Contact Person's Phone Number: +254 721958734

This contact information is provided to facilitate communication and access to the data used in our study. We assure you that the designated contact person is authorized to handle data access requests on behalf of our institution and will respond promptly to inquiries regarding data availability and access. Please feel free to contact us if you require any further information or clarification regarding the provided contact details.

5. We note that Figures 1 and 4 in your submission contain copyrighted images. All PLOS content is published under the Creative Commons Attribution License (CC BY 4.0), which means that the manuscript, images, and Supporting Information files will be freely available online, and any third party is permitted to access, download, copy, distribute, and use these materials in any way, even commercially, with proper attribution. For more information, see our copyright guidelines: http://journals.plos.org/plosone/s/licenses-and-copyright. We require you to either (1) present written permission from the copyright holder to publish these figures specifically under the CC BY 4.0 license, or (2) remove the figures from your submission.

Response: 

We respectfully assert that the images included in Figures 1 and 4 are original creations produced by our research team and do not breach any copyrights. As per your guidelines, we assure you that our manuscript, including the images in question, adheres to this licensing agreement. These images have not been previously published or distributed elsewhere. We appreciate the thorough review and attention to detail by the editorial team. We are confident that the inclusion of Figures 1 and 4 contributes significantly to the scientific merit and integrity of our manuscript.

6. We note that Figure 2 in your submission contain [map/satellite] images which may be copyrighted.

Response:

We appreciate the meticulous review conducted by the editorial team and the opportunity to address the concerns regarding the inclusion of map/satellite images in Figure 2. We wish to clarify that the image presented in Figure 2 is an original artwork/drawing created by our research team. The drawing was produced in-house specifically for this research project and does not contain any copyrighted materials or third-party content.

7. Please reduce the words in the abstract to 250 words.

Response: 

We have revised the abstract to meet the suggestion of reducing the word count. The revised abstract now contains fewer than 250 words while still effectively summarizing the key findings of the study.

8. I don't see the knowledge gap in the introduction.

Response:

We appreciate the reviewer's diligence in assessing the clarity of our manuscript. We've taken note of the comment regarding the identification of the knowledge gap and have since revised the Introduction section to provide a more detailed description of the gap our study seeks to address. Our study aims to bridge the existing research gap surrounding the functionality, usability, and maintenance of handwashing stations in public settings amid the COVID-19 pandemic. By conducting a comprehensive observational study, we aimed to shed light on the operational challenges and user perceptions associated with these facilities, particularly in the context of the pandemic response efforts. Through our findings, we endeavour to contribute valuable insights that can inform the optimization of hand hygiene practices and the implementation of effective public health interventions. We believe that these revisions adequately address the reviewer's concern and enhance the clarity of our manuscript. This modification is located in the last four paragraphs of the Introduction section.

9. I think lines 81-91 on page 4 should be put in the method, also the line…..

Response:

Thank you for your suggestion regarding the relocation of the paragraph from lines 81-91 on page 4 to the Methods section. We agree that this information is integral to understanding the methodology of our study. We have incorporated this paragraph into the Methods section under the subheading titled "Study setting and Population".

10. I suggest removing the sampling calculation to supplementary material

Response:

We appreciate your feedback regarding the inclusion of the sampling calculation in the main manuscript. Upon careful consideration, we agree that moving the sampling calculation to the supplementary material will improve the flow and readability of the main text. We have relocated the sampling calculation to the supplementary material, providing a clear reference in the main text to facilitate easy access for readers who wish to review the details. This adjustment will streamline the presentation of the study methods while ensuring that essential details remain accessible to interested readers.

11. Line 199-202 -> Why is the font size different?

Response:

Thank you for bringing attention to the discrepancy in font size. We have ensured that the font size is uniform throughout the entire document by standardizing it to the appropriate size as per the journal's guidelines.

12. Line 246-249 -> only one asterisk for two footnotes?

Response:

Thank you for bringing this to our attention. We appreciate your careful review of our manuscript. We have revised the Accessibility of Handwashing Stations section to address the issue of the footnotes. Each footnote now has its corresponding asterisk, as follows:

"* Accessibility was defined as HWS that are easily visible (spotted), conveniently reached by users and had visible nudges/IEC materials. 

** Only accessible to users immediately near the facility as other objects, such as market stalls, make the facility less visible."

13. Line 259-260 -> It is strange that the quote appears here without any explanation/narration before.

Response:

We appreciate your valuable feedback regarding the incorporation of quotes within the manuscript. In response to your concern about the quote appearing without preceding explanation or narration, we have revised the section on "Accessibility of Handwashing Stations in Public Spaces" to seamlessly integrate the quote within the context of the discussion. As suggested, we have introduced the quote immediately after presenting the findings related to the accessibility of handwashing stations.

14. Table 3 -> Please report statistical results properly.

Response:

Thank you for your feedback. We have revised Table 3 to present the association between washing hands using the proper technique and the presence of information, education, and communication (IEC) materials on handwashing stations more clearly. In the updated table, we have split the merged cells regarding the availability of IEC materials, presenting them separately under "Yes" and "No" categories. The total column is now standalone. Furthermore, we have removed the information on the Chi-square test from the last row and placed it after the footnote, providing a cleaner presentation of the statistical results.

15. I think the 5 tables (and some of them are big tables) are too much. Please consider reducing them and putting some in the supplementary.

Response:

Thank you for your suggestion regarding the number of tables included in the manuscript. We have carefully considered your feedback and have made adjustments accordingly. Tables 1, 2, 3, and 4 have been relocated to the supplementary material, as suggested, to streamline the main manuscript and improve its readability.

16. There should be a conclusion section. Please make the last paragraph in the discussion as your conclusion section.

Response:

Thank you for your feedback. We have revised the manuscript accordingly. The last paragraph in the discussion section now serves as the conclusion section, encapsulating the key findings and implications of the study.

Reviewer 1: 

Thank you for the opportunity to review the manuscript titled: Lessons from the deployment and management of public handwashing stations in response to the COVID-19 pandemic in Kenya: A Cross-sectional, observational study.

The authors sought to assess the functionality, usability and accessibility of rolled public handwashing stations in a COVID-19 pandemic context to improve their operation and maintenance and to inform strategies for future rollouts. The authors referenced multiple complementary methodologies and primarily found that the distribution of public handwashing facilities can have a limited lifetime, highlighting some implications of their findings. On the whole the manuscript was an interesting read and of value to PLoS One. There are some opportunities to bolster the manuscript. The main feedback is that additional clarification of the methods/measures and alignment of the results per the listed methodologies is sorely needed.

Here are my detailed comments and suggestions:

1. Abstract

a) Line 21: clarify that financial support was also provided.

Response:

Thank you for your valuable feedback. We have revised line 21 to now indicate that there were caretakers who were supporte financially to operate and manage the handwashing stations. 

b) Include other results- qualitative and exit surveys. Quantitative data were collected through spot checks, caretaker surveys, user observations and user-exit interviews; whereas qualitative data were collected using key informant interviews, focus group discussions and in-depth interviews.

Response:

Thank you for your valuable feedback. We have revised the abstract to include additional results from the qualitative surveys. It now reads, 'Qualitative data highlighted challenges such as inadequate signage, limited availability of soap and water, and issues related to the physical accessibility of the stations, particularly for individuals with disabilities.'

2. Introduction

a) 89: Include that fabricated stands, soap and soap holders were also a part of phase III.

Response: 

We appreciate your attention to detail and believe that this clarification strengthens the description of the intervention in phase 3. It now clearly states, 'In this final phase, 1558 brightly coloured 100 litre-capacity HWS with wastewater drainage pipes, fabricated stands, soap and soap holders, and IEC posters were distributed.' Please note that because of the editor's comments we moved this information to the Methods section under study setting and population. Please note that this paragraph has been moved to the Methods section under the 'Study setting and population' sub-heading.

b) 98: define what it meant by (O&M) and the specific tasks the caretakers were to carry out. It is unclear how soap and water were provided.

Response: 

In response to this query, we acknowledge the need for clarification regarding the term (O&M), which stands for operation and maintenance. In the context of our study, operation and maintenance encompassed a range of responsibilities assigned to the handwashing station caretakers. These tasks included ensuring the availability of soap and water, regular cleaning of the handwashing stations, refilling water containers, and restocking soap as needed. As for the provision of soap and water, these essential supplies were typically provided by the caretakers themselves or by individuals in the community. 

It now reads, 'Each handwashing station (HWS) was assigned a caretaker responsible for its operation and maintenance. In the context of our study, operation and maintenance encompassed a range of responsibilities assigned to the handwashing station caretakers. These tasks included ensuring the availability of soap and water, regular cleaning of the handwashing stations, refilling water containers, and restocking soap as needed. Regarding the provision of soap and water, guidelines were established during deployment, emphasizing the responsibility of HWS recipients to regularly refill water and replenish soap for handwashing once the initial supply of 800mg provided by coalition partners was depleted. This task was undertaken either by the caretakers or other community members. The caretakers fell into two categories: volunteers, typically local business owners, or paid individuals. Paid caretakers, predominantly stationed in informal settlements, were contracted by non-governmental organizations affiliated with the National Business Compact Coalition (NBCC) for periods ranging from 3 to 6 months, coinciding with the peak distribution of HWS and COVID-19 caseload. Their role was to remain on standby, positioned next to or in close proximity to the HWS, to ensure optimal operation and maintenance. On the other hand, unpaid caretakers were volunteers residing near the HWS, often near their businesses such as stalls, shops, supermarkets, markets, or transport hubs. These caretakers, presumed to be respected members of the community, were expected to assume responsibility for the operation and maintenance of the HWS without formal compensation.' Please note that this paragraph has been moved to the Methods section under the 'Study setting and population' sub-heading.

3. Methods

a) 111: 

---

## [Decision Letter · Decision Letter 1]

19 Apr 2024

Lessons from the deployment and management of public handwashing stations in response to the COVID-19 pandemic in Kenya: A cross-sectional, observational study

PONE-D-23-25273R1

Dear Dr. Munai,

We’re pleased to inform you that your manuscript has been judged scientifically suitable for publication and will be formally accepted for publication once it meets all outstanding technical requirements.

Kind regards,

D. Daniel, Ph.D.

Academic Editor

PLOS ONE

Additional Editor Comments (optional):

I think all comments from Reviewers and me have been addressed well. Thank you

Reviewers' comments:

Reviewer's Responses to Questions

**Comments to the Author**

1. If the authors have adequately addressed your comments raised in a previous round of review and you feel that this manuscript is now acceptable for publication, you may indicate that here to bypass the “Comments to the Author” section, enter your conflict of interest statement in the “Confidential to Editor” section, and submit your "Accept" recommendation.

Reviewer #2: All comments have been addressed

2. Is the manuscript technically sound, and do the data support the conclusions?

Reviewer #2: Yes

3. Has the statistical analysis been performed appropriately and rigorously? 

Reviewer #2: Yes

4. Have the authors made all data underlying the findings in their manuscript fully available?

Reviewer #2: Yes

5. Is the manuscript presented in an intelligible fashion and written in standard English?

Reviewer #2: Yes

6. Review Comments to the Author

Reviewer #2: Dear Authors

Thank you for revising the manuscript based on the reviewers' comments. The manuscript now are very clear stating the research gap and provide more concise methods and results sections. I have seen the inclusion of existing theory to discuss the findings.

7. PLOS authors have the option to publish the peer review history of their article (what does this mean?). If published, this will include your full peer review and any attached files.

Reviewer #2: **Yes: **Ni Made Utami Dwipayanti
